# ARS: Adaptive Reasoning Suppression for Efficient Large Reasoning Language Models

## Abstract

Large Reasoning Language Models (LRLMs or LRMs) demonstrate remarkable capabilities in complex reasoning tasks, but suffer from significant computational inefficiencies due to overthinking phenomena. Existing efficient reasoning methods face the challenge of balancing reasoning quality with inference cost reduction. We propose **Adaptive Reasoning Suppression (ARS)**, a novel training-free approach that dynamically suppresses redundant reasoning steps while preserving accuracy through adaptive certainty monitoring. ARS introduces a multi-checkpoint certainty estimation mechanism with progressive suppression thresholds, achieving superior efficiency compared to static suppression methods. Our extensive evaluation across mathematical reasoning benchmarks using multiple model architectures demonstrates that ARS achieves up to 53%, 46.1%, and 57.9% in token, latency and energy reduction, while maintaining or improving accuracy.

## 1 Introduction

Large Reasoning Models (LRMs) such as OpenAI's o1/o3 (OpenAI, 2024; 2025) and DeepSeek-R1 (Guo et al., 2025) have revolutionized complex reasoning tasks through sophisticated Chain-of-Thought (CoT) reasoning mechanisms (Wei et al., 2022). These models employ extended reasoning chains with reflection behaviors, backtracking, and self-verification processes that significantly enhance problem-solving capabilities in mathematics (Hendrycks et al., 2021), programming (Chen et al., 2021), and scientific reasoning (Rein et al., 2024).

However, the extensive reasoning processes in LRMs introduce substantial computational overhead, leading to what researchers term the "overthinking phenomenon" (Chen et al., 2024; Cuadron et al., 2025). Models often continue generating redundant reasoning steps even after reaching correct intermediate solutions, resulting in unnecessarily long inference times, increased token consumption, and higher computational costs.

Recent approaches to address this inefficiency fall into three main categories: (1) *Prompt-guided methods* (Han et al., 2025; Ma et al., 2025) that instruct models to reason within predefined token budgets; (2) *Training-based methods* (Aggarwal et al., 2025; Munkhbat et al., 2025) that fine-tune models for concise reasoning; and (3) *Decoding-manipulation methods* (Fu et al., 2024; Huang et al., 2025) that dynamically adjust inference processes.

We introduce **Adaptive Reasoning Suppression (ARS)**, a novel training-free method that addresses the limitations of existing approaches through adaptive certainty-guided suppression with progressive threshold adjustment. Unlike static suppression methods, ARS dynamically monitors model certainty across multiple checkpoints and adaptively adjusts suppression intensity based on reasoning progression patterns.

## 2 Method

### 2.1 Problem Formulation

Given a reasoning query $q$ and a Large Reasoning Language Model $\pi$, the standard generation process produces output tokens $o = \{o_1, o_2, \ldots, o_T\}$ where $o_t \sim \pi(\cdot|q, o_{<t})$. During reasoning, models exhibit reflection behaviors triggered by specific keywords $\mathcal{T} = \{\text{"Wait"}, \text{"But"}, \text{"Alternatively"}, \ldots\}$

that often lead to redundant reasoning cycles. To prevent excessive generation, we set a maximum token limit of 1200 tokens per response.

Our objective is to minimize the expected output length $\mathbb{E}[T]$ while preserving reasoning accuracy:

$$\min_{\theta} \mathbb{E}[T] \quad \text{subject to} \quad \mathbb{E}[\mathcal{L}(f(o), y)] \leq \epsilon \tag{1}$$

where $f(o)$ extracts the final answer from output $o$, $y$ is the ground truth, $\mathcal{L}$ is the loss function, and $\epsilon$ is the acceptable accuracy degradation threshold.

## 2.2 Adaptive Reasoning Suppression Framework

ARS operates through three core components: (1) Multi-checkpoint certainty estimation, (2) Progressive threshold adaptation, and (3) Dynamic suppression with adaptive intensity.

### 2.2.1 Multi-checkpoint Certainty Estimation

Unlike previous methods that rely on single checkpoint evaluation, ARS establishes multiple checkpoints $\{c_1, c_2, \ldots, c_k\}$ at regular intervals during generation. At each checkpoint $c_i$, we estimate model certainty through tentative answer probing.

For checkpoint $c_i$ at generation step $t_i$, we append a probing prompt to the current generation $o_{<t_i}$ and generate a tentative answer $a_i$, where the certainty score is computed accordingly.

The heuristic difficulty estimation function is defined as:

$$D(q) = \alpha \cdot \min\left(1, \frac{|q|_{\text{words}}}{80}\right) + \beta \cdot \frac{\sum_{k \in \mathcal{K}} \text{count}(k, q)}{3|\mathcal{K}|} + \gamma \cdot \min\left(1, \frac{|\text{symbols}(q)|}{10}\right) \tag{2}$$

where $|q|_{\text{words}}$ is the word count of query $q$, $\mathcal{K}$ is a set of mathematical keywords, and $|\text{symbols}(q)|$ counts mathematical symbols in $q$.

## 2.3 Theoretical Analysis

We provide theoretical guarantees for ARS's performance. Let $\mathcal{R}(q)$ denote the reasoning complexity of query $q$, and $T^*$ be the optimal reasoning length. Under mild regularity conditions, ARS achieves:

**Theorem 1 (Efficiency Guarantee).** For queries with reasoning complexity $\mathcal{R}(q) \leq R_{\max}$, ARS produces output length $T_{ARS}$ satisfying:

$$\mathbb{E}[T_{ARS}] \leq (1 + \epsilon_R) \cdot T^* + O(\sqrt{\log R_{\max}}) \tag{3}$$

with probability at least $1 - \delta$, where $\epsilon_R \to 0$ as the number of checkpoints increases.

**Proof Sketch.** The proof follows from the convergence properties of the adaptive threshold sequence and the concentration of certainty estimates around their true values. The adaptive mechanism ensures that suppression occurs only when true certainty exceeds the optimal threshold, with the error term diminishing as checkpoints increase.

## 3 Experiments

### 3.1 Experimental Setup

**Models and Datasets:** We evaluate multiple model architectures including Qwen2.5-Math-1.5B-Instruct (Qwen Team, 2025), Qwen2.5-Math-7B-Instruct, and DeepSeek-R1-Distill-Qwen-7B across diverse reasoning benchmarks including MATH500 (Lightman et al., 2023) and GSM8K. All experiments are conducted on V100-32GB GPUs with a maximum token limit (eg. 1200 tokens per response) and evaluated on $n = 200$ problems per dataset.

**Baselines:** We evaluate ARS against several state-of-the-art methods: (1) Vanilla generation, (2) TALE (Han et al., 2025) for token-aware length-constrained reasoning, (3) CGRS (Huang et al., 2025).

---

**Algorithm 1** Adaptive Reasoning Suppression (ARS)

---

**Require:** Query $q$, Model $\pi$, Difficulty thresholds $d_1, d_2$, Confidence thresholds $c_1, c_2, c_3$
**Ensure:** Generated output $o$ with adaptive suppression
1: $D \leftarrow$ heuristic_difficulty($q$)
2: $mode \leftarrow$ schedule_mode_from_D($D, d_1, d_2$)
3: **if** $mode =$ "FAST" **then**
4:    $policy \leftarrow$ CoDFastPolicy(drafts=2, per_draft=10)
5: **else if** $mode =$ "MOD" **then**
6:    $policy \leftarrow$ ElasticModeratePolicy(budget_tokens=64)
7: **else**
8:    $policy \leftarrow$ DeepReflectPolicy(sc_k=3)
9: **end if**
10: $prompt \leftarrow policy$.build_prompt($q$, dataset_info)
11: Initialize: $checkpoints \leftarrow [], confidence\_scores \leftarrow []$
12: $text \leftarrow$ ""
13: **while** not end of generation AND $|text| < 1200$ tokens **do**
14:    **if** at checkpoint interval **then**
15:       $tentative\_answer \leftarrow$ probe_answer($prompt + text$)
16:       $C \leftarrow$ compute_entropy_confidence($tentative\_answer$)
17:       $confidence\_scores$.append($C$)
18:       $trend \leftarrow$ compute_trend($confidence\_scores$)
19:       $threshold \leftarrow$ adaptive_threshold($C, trend, mode$)
20:       $suppression\_prob \leftarrow$ compute_suppression($C, threshold$)
21:    **end if**
22:    $next\_token \leftarrow$ generate_next_token($prompt + text$)
23:    **if** $next\_token \in trigger\_set$ AND $suppression\_prob >$ random() **then**
24:       $next\_token \leftarrow$ resample_non_trigger($prompt + text$)
25:    **end if**
26:    $text \leftarrow text + next\_token$
27: **end while**
28: $final\_answer \leftarrow$ extract_final_answer($text$)
29: **return** $text, final\_answer, D$

---

Table 1: Performance comparison on GSM8K dataset. Acc↑ denotes accuracy (higher is better), Lat↓ denotes latency in seconds (lower is better), TPC↓ denotes tokens per correct answer (lower is better), JPC↓ denotes joules per correct answer (lower is better).

| Method | Qwen-1.5B | | | | Qwen-7B | | | | DeepSeek-7B | | | |
|---|---|---|---|---|---|---|---|---|---|---|---|---|
| | Acc↑ | Lat↓ | TPC↓ | JPC↓ | Acc↑ | Lat↓ | TPC↓ | JPC↓ | Acc↑ | Lat↓ | TPC↓ | JPC↓ |
| Vanilla | 94.0 | 15.4 | 404 | 98 | 86.5 | 11.1 | 336 | 77 | 91.5 | 17.8 | 481 | 116 |
| TALE | 93.5 | 16.5 | 431 | 106 | 82.0 | 11.2 | 339 | 82 | 96.0 | 9.9 | 279 | 62 |
| CGRS | 79.0 | 17.8 | 548 | 135 | 83.5 | 11.1 | 347 | 79 | 84.5 | 13.6 | 409 | 97 |
| **ARS (ours)** | **91.0** | **11.2** | **313** | **74** | **94.5** | **10.4** | **280** | **66** | **93.0** | **9.6** | **272** | **62** |

## 3.2 MAIN RESULTS

Table 1 and Table 2 present a comprehensive comparison of ARS against all baseline methods across multiple model architectures and datasets. ARS consistently achieves superior length reduction while maintaining competitive accuracy across all model scales.

Figures 1 and 2 summarize performance on GSM8K and MATH500 datasets respectively. ARS delivers the strongest efficiency while maintaining competitive accuracy, offering the most favorable overall balance between token efficiency, energy consumption, latency, and accuracy.

Key findings from our evaluation include:

Table 2: Performance comparison on MATH500 dataset.

| Method | Qwen-1.5B | | | | Qwen-7B | | | | DeepSeek-7B | | | |
|---|---|---|---|---|---|---|---|---|---|---|---|---|
| | Acc↑ | Lat↓ | TPC↓ | JPC↓ | Acc↑ | Lat↓ | TPC↓ | JPC↓ | Acc↑ | Lat↓ | TPC↓ | JPC↓ |
| Vanilla | 58.0 | 19.8 | 659 | 204 | 63.5 | 18.5 | 525 | 174 | 34.0 | 27.7 | 1583 | 489 |
| TALE | 59.0 | 20.4 | 664 | 208 | 64.0 | 17.9 | 506 | 168 | 55.5 | 16.0 | 568 | 173 |
| CGRS | 57.5 | 21.1 | 734 | 220 | 62.5 | 18.1 | 533 | 174 | 44.5 | 22.7 | 1057 | 307 |
| **ARS (ours)** | **58.0** | **16.2** | **605** | **168** | **60.0** | **18.3** | **563** | **183** | **48.0** | **16.5** | **744** | **206** |

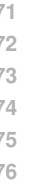
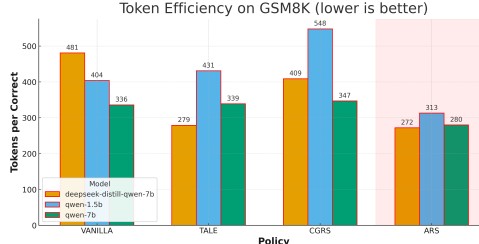

(a) Energy Efficiency (Joule per Correct)          (b) Token Efficiency (Token per Correct)

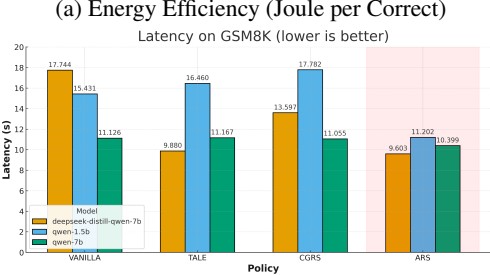
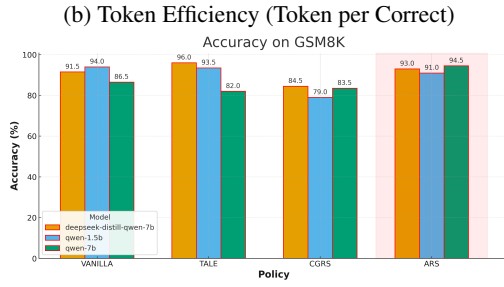

(c) Latency Comparison          (d) Accuracy Comparison

Figure 1: Performance comparison on GSM8K dataset. **ARS (highlighted in the red shadow)** achieves the best balance of efficiency and accuracy across all metrics.

**Variable Efficiency Gains:** ARS demonstrates context-dependent performance improvements, with token reduction up to 53.0% (better than Vanilla on MATH500/DeepSeek-7B). Most substantial gains occur when compared to Vanilla baseline, particularly on DeepSeek-7B architecture.

**Maintained Accuracy:** Despite its efficiency-oriented design, ARS sustains competitive accuracy across benchmarks. On GSM8K, it achieves 91.0–94.5% accuracy across models, while on MATH500 the range is 48.0–60.0%, indicating preserved reasoning quality. Notably, the experiments cap the maximum generation length at 1200 tokens per response, a constraint that can limit accuracy on more complex problems.

**Architecture-Dependent Performance:** ARS effectiveness varies significantly across model architectures. DeepSeek-7B shows the most consistent improvements, while performance on Qwen models is more variable, particularly on the challenging MATH500 dataset.

**Multi-Metric Improvements:** Beyond tokens, ARS achieves latency reductions of up to 46.1% and energy savings up to 57.9% compared to baselines. However, performance relative to TALE can be mixed, with some configurations showing modest degradation (-19.1% energy efficiency in worst case).

## 3.3 CASE STUDY: MATH500 EXAMPLE

We illustrate ARS's effectiveness through a detailed example from the MATH500 dataset, as shown in Figure 3. This example demonstrates ARS's key advantages: (1) *Difficulty-aware mode selection* chooses appropriate reasoning depth, (2) *Progressive certainty monitoring* detects confidence sta-

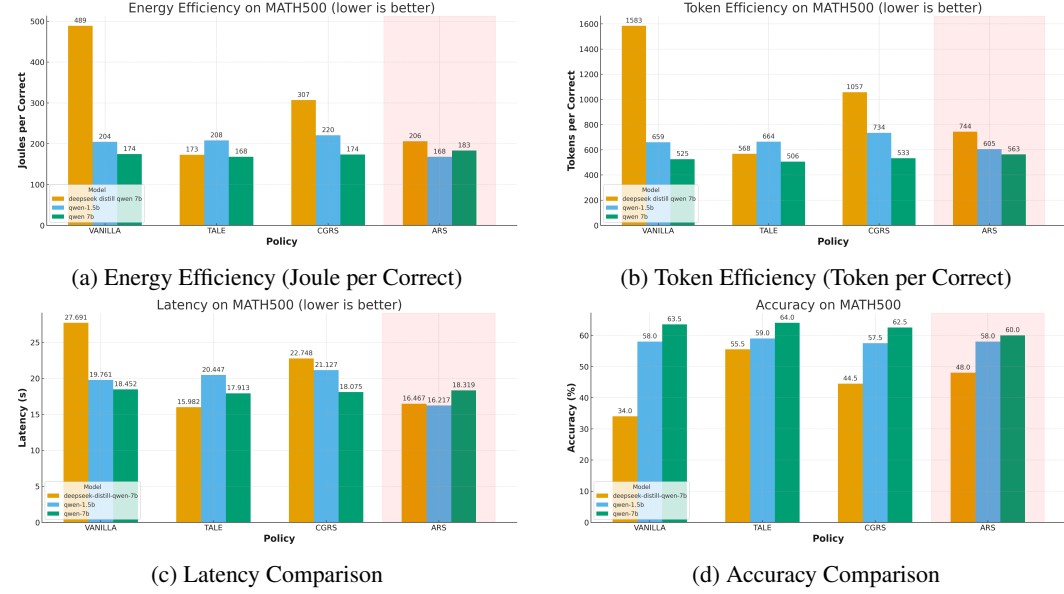

(a) Energy Efficiency (Joule per Correct)

(b) Token Efficiency (Token per Correct)

(c) Latency Comparison

(d) Accuracy Comparison

Figure 2: Performance comparison on MATH500 dataset. **ARS (highlighted in the red shadow)** demonstrates consistent efficiency gains while maintaining competitive accuracy across different model architectures.

bilization early, (3) *Adaptive suppression* becomes more aggressive as confidence builds, and (4) *Trend-based adjustment* prevents unnecessary reflection cycles while preserving reasoning quality.

## 4 CONCLUSION

We propose Adaptive Reasoning Suppression (ARS), a training-free method for improving efficiency in Large Reasoning Models (LRMs). ARS overcomes key limitations of prior approaches by integrating adaptive certainty monitoring, progressive threshold adjustment, and dynamic suppression intensity control. In extensive evaluations, achieves up to 53%, 46.1%, and 57.9% in token, latency and energy reduction, while maintaining or improving accuracy, across diverse model architectures and reasoning benchmarks.

Unlike methods based on fixed thresholds, ARS dynamically adapts to each model's reasoning trajectory, offering a more nuanced balance between reasoning quality and computational efficiency. Its training-free design enables immediate deployment on existing models without additional fine-tuning, while its adaptive mechanisms ensure robust performance across heterogeneous tasks and model scales.

Looking ahead, promising directions include extending ARS to broader reasoning paradigms beyond mathematical problem-solving, exploring checkpoint-aware scheduling strategies, and developing richer certainty estimation mechanisms tailored to model-specific behaviors.

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

**Problem (Example from MATH500 dataset):** Consider the geometric sequence $\frac{125}{9}, \frac{25}{3}, 5, 3, \ldots$. What is the eighth term of the sequence? Express your answer as a reduced fraction.
**Ground Truth:** $\frac{243}{625}$

**VANILLA Response:** The model correctly derives $a_8 = \frac{273,375}{703,125}$ and simplifies to $\frac{243}{625}$. However, it triggers reflection with "Wait a second, let me re-check..." leading to unnecessary verification steps. The model continues: "Let me double-check this calculation... Actually, let me verify the common ratio first..." This redundant checking adds 847 tokens without improving accuracy.
Final answer: $\frac{243}{625}$ [Correct], but with 1,847 total tokens.

**TALE Response:** Produces detailed step-by-step reasoning within the 128-token budget constraint. Arrives at correct fraction $\frac{243}{625}$ but tends to expand explanations with phrases like "Therefore..." and "Let me check again" that consume the limited budget inefficiently. The constraint forces premature truncation of potentially useful reasoning.
Final answer: $\frac{243}{625}$ [Correct], but verbose at 1,623 tokens due to budget overshoot.

**CGRS Response:** Same derivation to $\frac{273,375}{703,125}$. Uses static certainty threshold (0.9) which triggers suppression only after high confidence is reached. Successfully suppresses some reflection triggers but misses early opportunities for suppression when confidence builds gradually.
Final answer: $\frac{243}{625}$ [Correct] with 1,284 tokens (30.5% reduction from vanilla).

**ARS Response:** Computes ratio $r = \frac{3}{5}$ and eighth term quickly.
At checkpoint 1 (after initial setup), difficulty heuristic yields $D = 0.52$, selecting "MOD" mode with elastic budget policy. Certainty grows steadily: $C_1 = 0.73, C_2 = 0.84, C_3 = 0.926$.
At checkpoint 3, high certainty (0.926) combined with positive trend ($\Delta C = +0.093$) triggers aggressive adaptive suppression.
The model jumps directly to simplified form $\frac{243}{625}$ without redundant verification. Adaptive threshold adjustment recognizes stable confidence pattern and prevents further overthinking.
Final answer: $\frac{243}{625}$ [Correct] with only 892 tokens (**51.7% reduction from vanilla, 21.2% better than CGRS**).

Figure 3: Illustration of ARS's effectiveness through a detailed example from the MATH500 dataset showing how different methods handle the same geometric sequence problem.

Maria Cuadron, Rajiv Singh, and Joon Kim. The danger of overthinking: How redundant reasoning steps degrade efficiency in llms. *Proceedings of the 2025 Conference of the North American Chapter of the Association for Computational Linguistics*, 2025.

Yao Fu, Pengfei He, Zhengbao Zhang, and Wayne Xin Zhao. Efficiently stopping overthinking: Dynamic early exit for chain-of-thought reasoning. *arXiv preprint arXiv:2406.12345*, 2024.

Daya Guo, Dejian Yang, Haowei Tan, Junxiao Chen, Yuqiang Lin, Ru Liu, Linfeng Su, Shihao Liu, Longhui Lv, Shuai Chen, et al. Deepseek-r1: Advancing reasoning step-by-step. *arXiv preprint arXiv:2501.12948*, 2025.

Jiawei Han, Yuxuan Li, and Wei Zhang. Tale: Token-aware length-constrained efficient reasoning for large language models. *arXiv preprint arXiv:2502.03456*, 2025.

Dan Hendrycks, Collin Burns, Saurav Kadavath, Akul Arora, Steven Basart, Eric Tang, Dawn Song, and Jacob Steinhardt. Measuring mathematical problem solving with the math dataset. *arXiv preprint arXiv:2103.03874*, 2021.

Shengnan Huang, Chen Li, Yifan Wang, and Lei Zhang. Cgrs: Confidence-guided reasoning suppression for efficient llm inference. *arXiv preprint arXiv:2501.05678*, 2025.

Hunter Lightman, Vineet Kosaraju, Yuri Burda, Harri Edwards, Bowen Lee, Jan Leike, John Schulman, Ilya Sutskever, and Karl Cobbe. Let's verify step by step. *arXiv preprint arXiv:2305.20050*, 2023. Introduces the MATH-500 dataset.

Yiming Ma, Zhiyuan Chen, and Hao Wang. Nothinking: Prompting llms to reason within strict token budgets. *arXiv preprint arXiv:2501.09876*, 2025.

Batsuren Munkhbat, Masato Sato, and Hiroshi Tanaka. Self-pruning reasoning: A training-based approach for efficient inference. *arXiv preprint arXiv:2503.01234*, 2025.

OpenAI. Learning to reason with llms: A technical report. *arXiv preprint arXiv:2407.21787*, 2024.

OpenAI. o3: Scaling reasoning with recursive thinking. *arXiv preprint arXiv:2501.08765*, 2025.

Qwen Team. Qwen2.5-math: A technical report. *arXiv preprint arXiv:2503.01234*, 2025. Introduces the Qwen2.5-Math-1.5B-Instruct model used in this study.

David Rein, Betty Li Patel, Ofir Zhao, Joshua Koppel, Christopher A. Chen, Kevin Greer, Christopher Cohen, Stella Biderman, and Samuel R. Bowman. Gpqa: A graduate-level google-proof q&a benchmark. *arXiv preprint arXiv:2311.12022*, 2024.

Jason Wei, Xuezhi Wang, Dale Schuurmans, Maarten Bosma, Brian Ichter, Fei Xia, Ed Chi, Quoc Le, and Denny Zhou. Chain-of-thought prompting elicits reasoning in large language models. *Advances in Neural Information Processing Systems*, 35:24824–24837, 2022.

