# OpenReview forum: "ARS Adaptive Reasoning Suppression for Efficient Large Reasoning Language Model"
_ICLR.cc/2026/Conference — ICLR 2026 Conference Desk Rejected Submission_

### Official Review · Reviewer_fxsE · 2025-10-31

**Soundness:** 1
**Presentation:** 1
**Contribution:** 1
**Rating:** 0
**Confidence:** 4

**Summary:**

This paper introduces Adaptive Reasoning Suppression (ARS), a training-free approach designed to improve the computational efficiency of Large Reasoning Language Models (LRMs). The main idea is to ease the overthinking phenomenon by dynamically suppresses the redundant reasoning steps. There are three coponents proposed: Multi-checkpoint certainty estimation, Progressive threshold adaptation and Dynamic suppression. The results demonstrate that ARS achieves efficiency gains while maintaining or improving accuracy.

**Strengths:**

- The "overthinking phenomenon" is a real and important issue.
- The core idea of being training-free is a practical strength.
- The concept of using adaptive thresholds based on trends is more nuanced than a static, fixed threshold (like CGRS's).

**Weaknesses:**

This paper is full of holes and missing details, putting it in no shape for publication. Here are some major concerns:
1. The method is not well explained:
  - What exactly is the compute_entropy_confidence function?
  - How is the adaptive_threshold determined? Is it a learned function or a heuristic?
  - Algorithm 1 contains many undefined functions (probe_answer, compute_trend, Policy functions, etc.).
  - The parameters ($\alpha$, $\beta$, $\gamma$) in Equation 2 are not explained or tested.
2. The experimental setup is vague:
  - How was energy measured? How were the baselines (TALE, CGRS) run? Hyperparameters and setup are missing.
  - There is a major contradiction in the token limit: the paper states a "maximum token limit" of 1200, but Table 2 shows a TPC of 1583 for a baseline. And this 1200 token limit also seems arbitrarily short for complex reasoning
3. The evaluation is thin:
  - Only two baselines are measured, but the introduction implies many other training-free methods exist.
  - Testing on only 200 data points, and only on math dataset is likely not statistically significant.
4. The theory is incomplete and somewhat meaningless:
  - The theoretical analysis (Section 2.3) relies on undefined terms and provides only a one-sentence "Proof Sketch" instead of a proper proof.

**Questions:**

See weaknesses

---

### Official Review · Reviewer_A2c7 · 2025-10-31

**Soundness:** 1
**Presentation:** 1
**Contribution:** 1
**Rating:** 0
**Confidence:** 5

**Summary:**

The paper proposes Adaptive Reasoning Suppression (ARS), a training-free method designed to improve the efficiency of large reasoning language models. ARS dynamically suppresses unnecessary reasoning through adaptive certainty monitoring, where the model establishes multiple checkpoints during generation, prompts for tentative answers, and evaluates their confidence scores to decide whether to continue or suppress further reasoning. Empirical analysis shows some improvements on metrics like LAT. while maintaining accuracy.

**Strengths:**

The paper is targeting a real-world problem of easing the overhead for long reasoning output trace.

**Weaknesses:**

I am very concerned about the writing quality of the paper. On a high level. The paper lacks too many details, both in methodology and experimental results. There is also no related work, background nor detailed explanation of the methods, making it difficult to understand.


Starting from section 2, what is “MULTI-CHECKPOINT” vs. “single checkpoint”.

How the equation 2 is designed? Why choose those configurations like “80”/”10” and design the formula in such a way. In the meantime, what is the mathematical keyword? Why counting “mathematical keywords” and “symbols” helped evaluating the confidence.

How is theorem 3 correct? Is there a proof instead of proof sketch. Even if it is correct, how does it guarantee the efficiency.

Prompting the model in each generation phase adds up the efficiency overhead, do you count it into the end-to-end efficiency analysis?

It lacks the recent prompt based efficient reasoning baselines i.e. CoD.

Chain of Draft: Thinking Faster by Writing Less

**Questions:**

See weakness

---

### Official Review · Reviewer_9Bwa · 2025-11-01

**Soundness:** 1
**Presentation:** 1
**Contribution:** 1
**Rating:** 0
**Confidence:** 5

**Summary:**

This paper proposes ARS, a training-free, truncation-based efficient reasoning method.

**Strengths:**

Efficient reasoning is an important topic under the era of LRM.

**Weaknesses:**

Respectfully, this paper is not in a state suitable for submission. In terms of presentation, it lacks proper formatting and key components, such as related work and an ablation study, which are crucial for supporting the claims.

Regarding novelty, the core idea of truncating reasoning traces at different intervals to assess whether the current reasoning trace is sufficient has already been explored in works such as HALT-COT, DEER, Answer Consistency, FlashThink, and likely many other CoT compression techniques. Unfortunately, none of these works are mentioned nor compared in the paper.

The experimental setup described in Section3.1 is also lacking in several critical areas, including but not limited to:

1. Qwen 2.5 instruct models are not LRMs, so they are nor appropriate for this task.
2. #1 makes the only LRM to be the DeepSeek distilled 7B one. Evaluating only one LRM is not enough.
2. Setting the answer length to 1200 tokens is far too low to be representative. This likely explains why the DeepSeek-Distilled 7B model performs worse than the non-reasoning model of the same family.
3. Evaluating only 200 questions from common datasets like MATH500 and GSM8K provides an overly limited perspective.
4. There is a lack of basic experimental settings information, such as temperature, number of runs, etc.

These are just a few of the many issues with the paper, but I believe they are significant enough to warrant a strong rejection. **My sincere recommendation to the authors is to ensure the work is more complete before submitting it for peer review.** As it stands, the paper feels unfinished even from a page-limit perspective (only 5 pages, with at least one page occupied by pseudo-code and a visualization of already presented tables). As authors, we often hope for thoughtful and engaging reviewers, but part of achieving that comes from submitting work that is fully developed so that the review resource can be better distributede.

**Questions:**

N/A

---

### Official Review · Reviewer_ZHz9 · 2025-11-04

**Soundness:** 1
**Presentation:** 1
**Contribution:** 1
**Rating:** 0
**Confidence:** 4

**Summary:**

This paper proposes Adaptive Reasoning Suppression (ARS), a training-free approach that applies multi-checkpoint certainty estimation to suppress the reasoning process. However, the submission is incomplete. Key sections such as Method and Experiments are only partially written, making it difficult to evaluate the paper’s contributions and merits.

**Strengths:**

The paper is currently incomplete, and therefore no clear strengths can be identified at this stage.

**Weaknesses:**

The paper is incomplete, which makes it difficult to understand the proposed methodology, as many essential details are missing. As a result, it is challenging to identify any clear scientific merit at this stage.

**Questions:**

n/a

---

### Note · Program_Chairs · 2026-01-17
**Submission Desk Rejected by Program Chairs**

The following references in this submission do not refer to real documents and/or have major errors in bibliographic information:

 Yao Fu, Pengfei He, Zhengbao Zhang, and Wayne Xin Zhao. Efficiently stopping overthinking: Dynamic early exit for chain-of-thought reasoning. arXiv preprint arXiv:2406.12345, 2024.